



# Evapotranspiration at four sites representing land-use and height gradient in the Eastern Ore Mountains (Germany)

Uta Moderow, Stefanie Fischer, Thomas Grünwald, Ronald Queck, Christian Bernhofer

TU Dresden, Faculty of Environmental Sciences, Institute of Hydrology and Meteorology, Chair of Meteorology, Germany

*Correspondence to*: Uta Moderow (uta.moderow@tu-dresden.de)

**Abstract.** Less is known about evapotranspiration (ET) along elevation gradients of low mountain ranges, especially with regard to different land uses and concerning long-term studies. We investigate ET of four sites of different land-uses along an elevation gradient of a low mountain range over eleven years (2008-2018) based on daily values. Three different ET estimates are inspected, which can give a reasonable range of ET. These estimates are ET based on the energy balance

residual (ET_residual), ET corrected for the energy balance closure gap (ET_corr) and ET not corrected for the energy balance closure gap (ET_uncorr). In general, ET_residual showed largest values and ET_uncorr showed lowest values with ET_corr in between. Average annual differences between ET_residual and ET_corr ranged between 111 mm $a^{-1}$ and 196 mm $a^{-1}$. Average annual differences between ET_uncorr and ET_corr ranged between 70 mm $a^{-1}$ and 167 mm $a^{-1}$. For two site years ET_corr was lower than ET_uncorr. This could be related to gap-filling. Differences between different estimates were

site-specific and related to the respective energy balance closure gap. Principal component analysis revealed similar dependency on driving variables for all three estimates and all sites. Given the influence of the energy balance closure gap on ET_uncorr and ET_residual, we recommend using ET_corr, but ET_residual can be still useful especially for sites with low vegetation, which rarely experience water stress. Comparison of two coniferous sites situated at different altitudes showed frequently larger values for the site located at a higher altitude. This might be a result of rainfall interception, which

however must be investigated at sub-daily timescale.

## 1 Introduction

Evapotranspiration (ET) is a key process in the earth-atmosphere system as it is an important component of the water balance and is related to the latent heat flux, a component of the energy balance at the earth surface, via the latent heat of evaporation and sublimation, respectively. Thus, it is a highly relevant process in the earth-atmosphere system concerning

the exchange of mass and energy.

However, measurements of ET are still a challenging task. The Eddy Covariance (EC) method has gained outstanding importance in measuring ET over the last 25 years (e.g. Goulden et al. 1996; Aubinet et al. 1999; Bernhofer and Vogt 1999; Wilson et al. 2001; Baldocchi 2003; Panin and Bernhofer 2008; Moderow et al. 2009; Aubinet et al. 2012, Baldocchi 2014).





The broad application of this technique in international networks, which address the exchange between the earth and the
atmosphere (FLUXNET, ICOS), reflects this but also the vast number[1] of publications using EC based ET-data.

However, we often encounter the problem of energy balance closure gap (e.g. Tsvang et al. 1991; Kanemasu et al. 1992; Wilson et al. 2002; Foken 2008; Franssen et al. 2010; Stoy et al. 2013; Gerken et al. 2018; McGloin et al. 2018) when assessing the surface fluxes of latent (LE) and sensible heat (H) of the energy balance at the earth surface by means of EC. Commonly the measured sum of LE and H is often smaller than the measured sum of the available energy (net radiation
minus ground heat flux minus heat storages changes). The measured components of the energy balance do not sum up to zero. An energy residual remains, often called energy balance closure gap. Most multi-site studies report an energy balance closure gap between 10% and 30%. (Wilson et al. 2002; Foken 2008; Franssen et al. 2010; Stoy et al. 2013). Only a few studies report a negligible energy balance closure gap (Heusinkveld et al. 2004; Mauder et al. 2007).

EC based ET is affected by the energy balance problem and, therefore, very likely underestimates the actual
evapotranspiration. Several methods have been proposed for a partition of the energy balance residual to LE and H in order to arrive at more reliable estimates of LE and hence ET (Blanken et al. 1997, Twine et al. 2000; Barr et al. 2012; Mauder et al. 2013; Charuchittipan et al. 2014; FLUXNET 2017; De Roo et al. 2018). There is an ongoing discussion on how the missed energy should be partition between H and LE (Mauder et al. 2013; Charuchittipan et al. 2014; De Roo et al. 2018; Mauder et al. 2018).

Different methods for partitioning the residual between H and LE inevitably produce different estimates of LE and hence ET. However, at least we need to know a reasonable range for ET. A reasonable upper border would be given by ET estimates based on LE determined as a residual of the energy balance (ET_residual) whereby all components of the energy balance are measured but not LE. This method has been frequently applied in different studies and compared to other estimates of ET (e.g. McNeil and Shuttleworth 1975; Amiro and Wuschke 1987; Adams et al. 1991;Twine et al. 2000;
Amiro 2009; Wohlfahrt et al. 2010, Barr et al. 2012; Spank et al. 2013; Mauder et al. 2018). ET_residual overestimates ET in relation to ET based on LE corrected for the energy balance closure gap (ET_corr) or not corrected for the missed energy (ET_uncorr) as reported by (e.g.) Twine et al. 2000; Barr et al. 2012; Gebler et al. 2015; Castellvi and Oliphant 2017; Perez-Priego et al. 2017 and Mauder et al. 2018). However, only a few studies published results for several years (Barr et al. 2012; Mauder et al. 2018).

The objective of this paper is to investigate differences and similarities and the possible range of these ET-estimates. It is focused on long-term ET of low mountain ranges along an elevation gradient, commonly less studied. It will address differences in ET due to different land uses and different altitude. The work is based on four sites of different land uses (coniferous forests, grassland, crop rotation) and of different altitudes of the Cluster of the Technische Universität Dresden

---

[1] approx. 23 000
https://scholar.google.com/scholar?hl=de&as_sdt=0%2C5&q=evapotranspiration+eddy+covariance&oq=Evapotranspiration+Eddy assessed 04.04.2020



for greenhouse gas and water fluxes (Moderow and Bernhofer 2014). The investigated period covers 11 years and daily
values are used for analyses.

## 2 Sites and Instrumentation

### 2.1 Site description

All four investigated sites of this study belong to the Cluster of the Technische Universität Dresden for greenhouse gas and
water fluxes (Moderow and Bernhofer 2014). Three of these sites are located about 15-25 km southwest of Dresden
(Germany; Fig. 1) in or close to the Tharandt forest (60 km²) in the lower region of the Eastern Ore Mountains
(Osterzgebirge). These sites are Anchor Station Tharandter Wald (DE-Tha, old spruce forest, EC-data since 1996),
Grillenburg (DE-Gri, grassland site, EC-data since 2003) and Klingenberg (DE-Kli; crop site, EC-data since 2005). The forth
site Oberbärenburg (DE-Obe, spruce forest, some EC-data since 1994) is situated at a higher altitude of the Ore Mountains
about 50 km south of Dresden (Germany). All four sites contribute to FLUXNET (https://daac.ornl.gov/cgi-
bin/dataset_lister.pl?p=9 assessed 29.04.2020). DE-Tha, DE-Gri and DE-Kli are ICOS sites (www. https://www.icos-cp.eu/
assessed 29.04.2020). Only complete years with similar instrumentation at all four sites are used in this paper. According to
the Köppen-Geiger climate classification, all sites belong to the climate type Cfb (warm temperate climate, fully humid,
warm summer; Kottek et al. 2006).

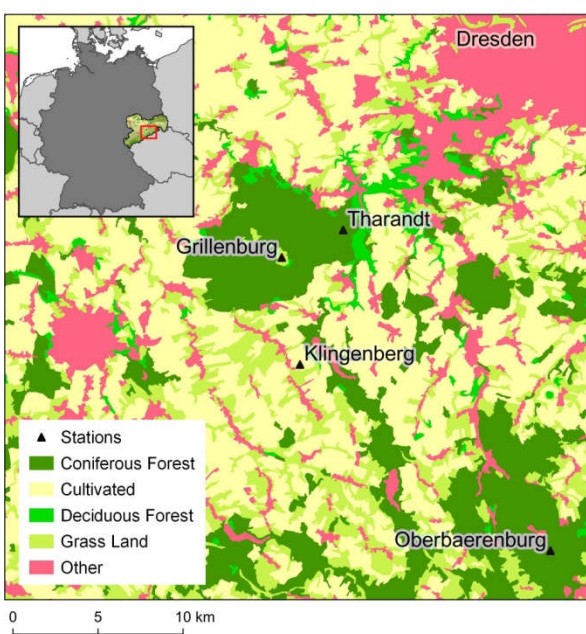

**Figure 1: Locations of the four investigated sites (Tharandt DE-Tha; Grillenburg DE-Gri; Klingenberg DE-Kli; Oberbärenburg DE-Obe). Map is based on freely available Geodata.**





An overview of general climatic conditions of all four sites during the investigated period (years 2008-2018) gives Fig. 2.
DE-Tha was the site with highest annual mean temperature (9.4 °C), followed by DE-Gri and DE-Kli (8.8 °C and 8.0 °C,
respectively) and DE-Obe being the coldest site (6.7 °C). Based on corrected numbers DE-Obe received most precipitation
with an average of 1176 mm a$^{-1}$ for the investigated period due to its location at a higher altitude. The mean annual
precipitation sums of the lower altitude sites are 906 mm a-1 (DE-Kli), 987 mm a$^{-1}$ (DE-Tha) and 1022 mm a$^{-1}$ (DE-Gri).
During the investigated period (years 2008-2018), the year 2010 was comparatively cool and wet (Fig. 2b and 2d) whereas
2018 was a very warm and extremely dry year, also in relation to available long-term records at DE-Tha (30 year average of
period 1981-2010: 8.4°C).

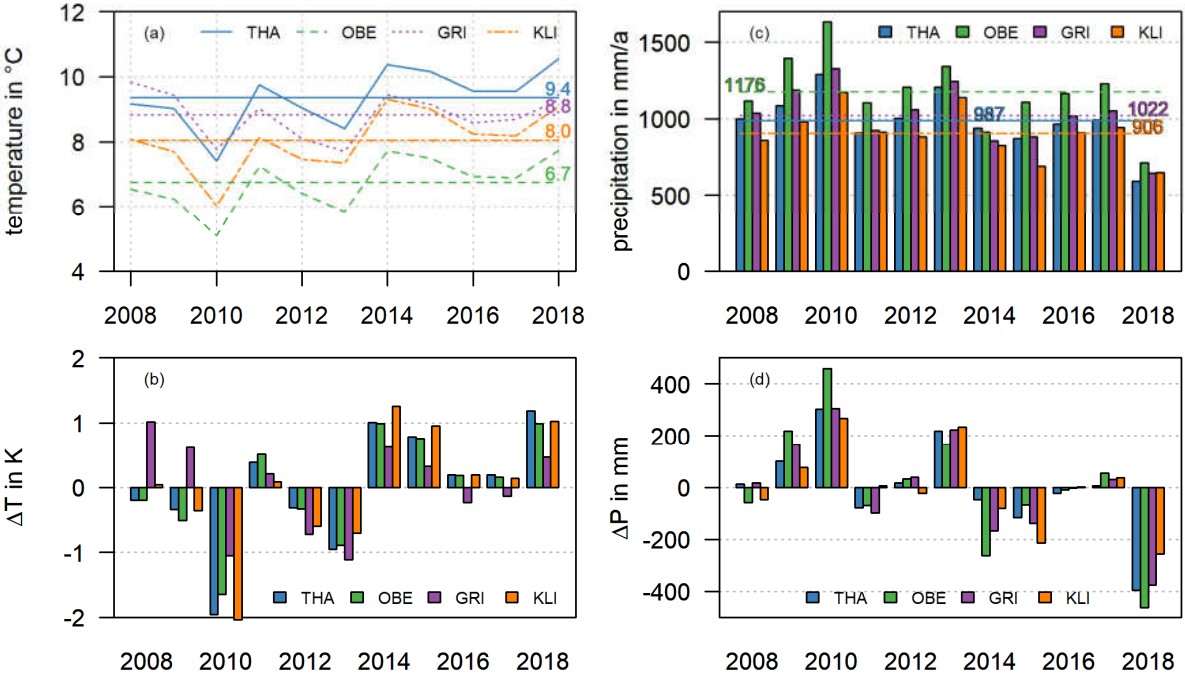

**Figure 2: Mean annual temperature (a) and precipitation sum (c) from 2008 to 2018 and respective averages of this period. Lower
panels show differences of each year's temperature (b) and precipitation (d) to the period's respective average value. THA stands
for Anchor Station Tharandter Wald (old spruce forest), GRI for Grillenburg (grassland site), KLI for Klingenberg (crop
rotation) and OBE for Oberbärenburg (spruce forest at a higher altitude of the Ore mountains).**

### 2.1.1 Old Spruce forest – Anchor station Tharandter Wald (DE-Tha)

The Anchor Station Tharandter Wald (50°58' N 13°34' E) is located in the eastern part of the Tharandt forest at an altitude of
385 m above sea level. It has an undulating terrain with a slope (2°) facing south. The Norway spruce stand (*Picea abies*)
was established by seeding in 1887 (Grünwald and Bernhofer 2007) and had mean tree height of 31 m in 2018. Maximum
plant area index is between 7 m$^2$ m$^{-2}$ and 8 m$^2$ m$^{-2}$. Norway spruce (*Picea abies*) is the dominating tree species (> 70 % of all
mature trees). Other species of trees are Scots Pine (*Pinus sylvestris*, 15 %), European Larch (*Larix decidua*, 10 %) and
subordinated different deciduous trees (3 %) like Silver Birch (*Betula pendula*), Norway Maple (*Acer platanoides*) and





Horse Chestnut (*Aesculus hippocastanum*) (Grünwald and Bernhofer 2007). Beech (*Fagus sylvatica*) underplanting took place in 1995 in some areas of the site. Those Beech trees reached a height about 3 to 4 m in 2007. During the investigated

period 2008-2018, the canopy was thinned in 2011 and 2016. The understorey is very sparse and dominated by Wavy Hair grass (*Deschampsia flexuosa*) or even absent in areas densely populated with spruce. Thus the canopy is characterized by a marked trunkspace.

The soil can be classified as loamy-skeletal podsol-brown earth (WRB: Dystric Cambisol) on rhyolite (Nebe and Wenk 1997) with a soil depth around 1 m and a main rooting zone of 35 cm. It has a high rock content which increases from the

upper soil layer to the lower soil layers (Schwärzel et al. 2009). The soil has a soil water content of 16 Vol% at field capacity and 7 Vol% at wilting point (Grünwald and Bernhofer 2007).

According to Mellmann et al. (2003) and Rebmann et al. (2005) the main (southwesterly) footprint of the flux measurements can be characterised as sufficiently homogeneous. The old spruce forest contributes at least 80% to the measured flux for 90% of the half-hourly flux measurements (Göckede et al. 2008).

### 110 2.1.2 Grassland site Grillenburg (DE-Gri)

The site Grillenburg (50° 57' N, 13°31 E) is a permanent pasture (mesophytic hay meadow). It is located at an altitude of 385 m at a large clearing of around 40 ha. Typical species of the grass cover are couch grass (*Agropyron repens*) meadow foxtail (*Alopecurus pratensis*), yarrow (*Achillea millefolium*), common sorrel (*Rumex acetosa*) and white clover (*Trifolium repens*).

The soil (Pseudogley, WRB: Stagnosol) is a deep soil (up to 1.35 m) of high silt content (at least 75% at all soil layers). The upper horizons are influenced by former ploughing. For the upper two horizons (23 cm) the wilting point is at 13 Vol% and the amount of water held between field capacity and wilting point is 30 Vol%

The permanent grassland has been unfertilized since 1987 and is extensively managed (one to three cuts per year for fodder and hay production, occasional cattle, sheep or horse grazing in autumn). According to cutting, the canopy height varies

over the year as well as the leaf area index. Maximum leaf area index before cutting is around 5-6 m² m$^{-2}$.

The grassland is surrounded by forest and the fetch is correspondingly restricted to 530 m (North), 250 m (West), 470 m (South) and 350 m (East), respectively.

### 2.1.3 Agricultural site Klingenberg (DE-Kli)

The agricultural site Klingenberg (50°54' N 13°31' E, 480 m above sea level) is located 4 km south of the Tharandt Forest at

a gentle slope facing south.

Different crops are grown here in a crop rotation cycle. Starting in 2008, the rotation cycle began with spring barley (*Hordeum vulgare*), followed by winter barley (*Hordeum vulgare*) in 2008/2009, rapeseed (*Brassica napus*) in 2009/2010, winter wheat (*Triticum aestivum*) 2010/2011 and maize (*Zea mays*) 2012. The crop rotation cylcle started again with spring barley (Hordeum vulgare) in 2013, followed by winter barley (*Hordeum vulgare*) in 2013/2014, rapeseed (*Brassica napus*)





2014/2015, winter wheat (*Triticum aestivum*) 2015/2016, followed by cover crop (cultivated radish, *Raphanus sativa brassica*) 2016/2017; spring wheat (*Hordeum vulgare*) 2017; cover crop (cultivated radish, *Raphanus sativa brassica*) 2017/2018 and maize (*Zea mays*) in 2018. Before 2016, DE-Kli represented fallow ground in periods between harvests and sowings of the subsequent crop. During this periods, only volunteer seedlings or weed species occurred.Due to crop rotation, canopy height and leaf area index varied from year to year and within a year.

The site is strongly influenced by management (low tillage, sowing, harvesting). Mineral fertilisation is applied several times every year. Organic fertilisation is also applied but not yearly. Herbicides are applied regularly several times every year.

The soil is a Gleysol, which is at least 80 cm deep. The upper horizon (0-20 cm) is influenced by ploughing and can be characterized as medium clayey loam. The adjacent horizon below is slightly sandy clay and clayey. Wilting point is at 26 Vol% and the field capacity has a soil water content of 41 Vol% for the upper 40 cm of the soil.

According to Spank et al. (2016) a sufficient fetch of at least 300 m can be assumed for the two main wind direction sectors (South-West and North-West).

### 2.1.3 Spruce forest Oberbärenburg (DE-Obe)

The site Oberbärenburg (50°47'N, 13°43') is located in the Eastern Ore Mountains at 735 m above sea level and slightly faces northeast. Oberbärenburg is situated in a region which was heavily damaged by smoke in the second half of the last

century (Queck 2004).

The age of the Norway spruce (*Picea abies*) stand was 55 years in 2010. The mean canopy height was 21 m in 2011. Maximum plant area index is between 7 $m^2$ $m^{-2}$ and 8 $m^2$ $m^{-2}$ (Moderow and Bernhofer 2014). Understorey is very sparse or even absent.

According to an adjacent forest research station of the Freestate Saxony, the soil is a podsol-brown earth (WRB: Dystric

Cambisol) on rhyolite. Loamy sand dominates until 0.25 m depth and deeper layers can be characterised as sandy loam.

During the investigated period 2008-2018, the canopy was thinned in 2016.

### 2.2 Instrumentation

We restrict the description of the instrumentation to the most relevant variables. Table 1 shows main instrumentation of the four investigated sites. Precipitation (P) measurements are based on weighing rain gauges at each site. In the case of DE-

Obe, precipitation measurements of a small gauged catchment (Rotherdbach, Zimmermann et al. 1999) is used, which is less than 1 km away of DE-Obe. P measurement of Rotherdbach is located at an altitude of 720 m. Comparison of precipitation measurement when P measurements of the catchment Rotherdbach and DE-Obe were both available revealed negligible differences. Precipitation measurements were corrected for wind errors according to Richter 1995.



Table 1: Measurement setups of the respective sites.

|  | DE-Tha | DE-Gri | DE-Kli | DE-Obe |
|---|---|---|---|---|
| Net radiation | CNR1[§] | CNR1[§] | CNR1[§] | CNR1[§] |
| Measurement height | 37 m | 1.5 m | 1.8 m | 30 m |
| Wind | GILL R3-50[%%] | GILL R3-50[%%] | Gill R3-50[%%;a] <br><br> Young 81000[&,a] | METEK USA-1[#] |
| Measurement height | 42 m | 3 m | 3.5 m | 30 m |
| Humidity (high frequency measurements) | Li-7000[$] | Li-7000[$] | Li-7000[$] | Li-7000[$] |
| Measurement height | 42 m | 3 m | 3.5 m | 30 m |
| Ground heat flux | PLE[&&] (2 plates) | PLE[&&] (2 plates) | PLE[&&] (2 plates) | PLE[&&] (2 plates) |
| Measurement height | -0.02 m | -0.02 m | -0.02 m | -0.02 m |
| Air temperature Relative humidity | HMP45[$$] | HMP45[$$] | HMP45[$$] | HMP45[$$] |
| Measurement height | 40 m | 2 m | 2 m | 30 m |
| Soil water content | TDR[§§] | TDR[§§] | TDR[§§] | TDR[§§,##] |
| Measurement height | -0.10 m | -0.10 m | -0.10 m | -0.10 m |
| Soil temperature | Thermocouple[%] | Thermocouple[%] | Thermocouple[%] | Thermocouple[%] |
| Measurement height | -0.02, 0.05 m | -0.02, 0.05 m | -0.02, 0.05 m | -0.02, 0.05 m |

[§]  Kipp & Zonen, Delft, The Netherlands
[§§]  IMKO, Ettlingen, Germany
[$]  LI-COR, Lincoln, Nebraska, USA
[$$]  Vaisala, Helsiniki, Finnland
[%]  Manufacturer not specified
[%%] Gill Instruments Ltd, Lamington, Hampshire, UK
[&]  R.M. Young Company, Traverse City, Michigan, USA
[&&] Laborelektronik Ing. Peter Leskowa, Austria
[#]   Metek, Elmshorn, Germany
[##]  since July 2018, Delta-T Devices, Burwell, Cambridge, UK
[a]  depending on availability



## 3. Methods

### 3.1. Evapotranspiration estimates

Evapotranspiration (ET) is part of the water balance,

$P = ET + R + \Delta S,$ (1)

where P denotes precipitation, ET evapotranspiration, R runoff and $\Delta S$ storage change. All terms are given in mm.

Via the latent heat of vaporization and sublimation L, respectively, ET is connected to the energy balance (Eq. 2 and Eq. 3) at the earth surface,

$ET = \frac{LE}{L}$ (2)

$Rn - G - J = H + LE,$ (3)

where Rn denotes net radiation, G ground heat flux, J heat storage changes, H sensible heat flux and LE latent heat flux. All energy balance components are given in W m$^{-2}$.

When assessing the energy balance at earth surface by means of Eddy-Covariance in general there is an energy imbalance, i.e. the left hand side of Eq.3 does not equal the right hand side of Eq. 3 (e.g. Foken et al. 2008, c.f. Introduction) and

commonly the sum of (H+LE) is smaller than the sum of (Rn – G – J). Consequently, measured ET is likely to be to small as its energy equivalent is underestimated. Therefore, ET values not corrected for this gap are likely to be smaller than the actual value. We will call these estimates ET_uncorr. ET estimates obtained by accounting for this gap will be called ET_corr.

In order to obtain daily values of ET_uncorr, half-hourly values of LE were averaged over the whole day (24 hour average).

As outlined before, different methods exist to correct for the missing energy (c.f. introductory part). LE correction follows the FLUXNET procedure (FLUXNET 2017), which is the current procedure applied within ICOS. This correction was applied to daily values of LE_uncorr. Daily values of ET_corr were then obtained by converting LE_corr to ET_corr using the latent heat of vaporization.

Besides ET_uncorr, ET_corr, the third estimate of ET is obtained via LE determined as a residual of the energy balance on a

daily basis.

$ET = \frac{LE}{L} = \frac{(Rn - G - J - H)}{L}$ (4)

where, L denotes latent heat of vaporization and sublimation, respectively, in J kg$^{-1}$. Estimates according to Eq. 4 will be referred to as ET_residual. Components of heat storage changes might be important when determining ET_residual for periods with snow and snowmelt (Amiro et al. 2009). Most studies report an overestimation of ET_residual with respect to

the chosen reference ET of the respective publication (Adams et al. 1991; van der Tol et al. 2003; Consoli et al. 2006; Wohlfahrt et al. 2010; Barr et al. 2012; Gebler et al. 2015; Castellvi and Oliphant et al. 2017; Perez-Priego et al. 2017; Mauder et al. 2018). We therefore assume that ET_residual is a reasonable estimate for an upper estimate of ET. However, differences to the chosen reference ET may vary with inspected time scale and season (Adams et al. 1991; Amiro 2009;



Perez-Priego et al. 2017) and with different measurement campaigns for the same site (Wohlfahrt et al. 2010). Furthermore,
results should be carefully reviewed as this methods piles up all errors of all other components of the energy balance in LE
(McNeil and Shuttleworth 1975; Barr et al. 2012) and hence ET.

### 3.2 Eddy – Covariance data

The sensible and latent heat fluxes have been measured using Eddy-Covariance (Aubinet et al. 2012). Raw data were
recorded at frequency of 20 or 25 Hz. EddyPro® version 6.2.0 (LI-COR, Lincoln, Nebraska, USA) was used for post-
processing to obtain half-hourly data of H, LE and ET. Following flux corrections have been implemented: Coordinate
rotation according to Wilczak et al. (2001) for 8 different wind sectors with a size of 45°, humidity correction of sonic
temperature (Dijk et al. 2004; correction for cross wind contamination was already implemented in the software of the used
sonic anemometers), correction for high frequency spectral losses (Fratini et al. 2012 and Horst and Lenschow 2009) as well
as low frequency spectral losses (Moncrieff et al. 2005). Ibrom et al. (2007) has been chosen in order to convert gas
concentrations into mixing ratios. For measurement heights and setups please refer to Table 1.

### 3.3 Heat storage changes of energy balance at the earth surface

Commonly, it is not possible to measure all components of the energy balance at the earth surface directly. Therefore, one
has to account for the different heat storage changes in the layer between the uppermost measurement height of an energy
balance component, commonly the height of the EC-system measuring latent and sensible heat flux above the canopy, and
the lowest measurements height of an energy balance component commonly the depth of the heat flux plate measuring the
ground heat flux (c.f. Oke 1987; Arya 2001).

Different heat storage changes contribute to the total heat storage change (Thom 1975, McCaughey 1985, Bernhofer et al.
2003; Oliphant et al. 2004, Moderow et al. 2009). Calculated heat storage changes of the energy balance equation included
following components (Eq. 5)

$$J = J_H + J_{LE} + J_C + J_{veg} + J_G ,$$ (5)

where $J_H$ and $J_{LE}$ denote sensible and latent heat storage changes in the canopy air layer, $J_C$ accounts for the energy fixed and
released by photosynthesis and respiration, respectively, and $J_{veg}$ denotes heat storage changes in the vegetation. $J_G$ accounts
for possible heat storage changes between the soil surface and the depth of the heat flux plate. All heat storage changes are
given in W m$^{-2}$, have been calculated on the basis of gap-filled half-hourly data and averaged over 24 hours in order to obtain
daily values. A description of the assessment of the different heat storage changes of Eq. 5 is given in Appendix A. $J_{veg}$ was
calculated for the two coniferous sites (DE-Tha, DE-Obe) but not the crop site (DE-Kli) and the grassland site (DE-Gri) as
sufficient data concerning fresh weight of biomass is missing. Furhermore, results of Eshonkoluv et al. (2019) and Jacobs et
al. (2008) indicated that $J_{veg}$ of low vegetation is be of minor importance compared to other heat storage change components.





### 3.4 Gap filling

Gap filling is inevitable in order to achieve yearly, seasonal and monthly budgets of LE and ET, respectively. Table 2 gives an overview of missing half-hourly data of main energy balance components before gap-filling. Statistic of H and LE relate to half-hourly data after post-processing using EddyPro.

Table 2: Percentage of missing half-hourly data of main energy balance components before gap-filling. Total number of half-
hourly values of the eleven year period 2008-2018: n=192864.

| Site | Rn | G$^\$$ | LE | H |
|------|------|------|------|------|
| DE-Tha | 1.2 | 0.3 | 3.3 | 1.8 |
| DE-Gri | 1.2 | 1.3 | 8.7 | 7.0 |
| DE-Kli | 7.0 | 12.6 | 20.3 | 10.7 |
| DE-Obe | 1.8 | 1.1 | 11.9 | 9.6 |

$^\$$Measured half-hourly values represent the average of two heat flux plates or the measurement of a single heat flux plate, depending on availability

Net radiation was gap-filled following Allen et al. (1994). Missing half-hourly values of G were filled with a moving
average over a time window of -/+ 7 days from a particular half hourly value. The calculation was only made if at least 5 half-hour values were available within the two weeks. This procedure was repeated until all gaps of G were filled. Half-hourly values of H and LE were gap-filled using the algorithm of Reichstein et al. (2005) and the corresponding online tool (REddyProcWeb; https://www.bgc-jena.mpg.de/bgi/index.php/Services/REddyProcWeb viewed 15. April 2020).

Half-hourly data, needed for the calculation of the heat storage changes (Eq. 3 and Eq. 5), were also gap-filled. These data
included net ecosystem exchange of $CO_2$ (NEE), soil temperatures, air temperature, relative humidity and water vapour pressure deficit. Most gaps could be filled using the algorithm of Reichstein et al. (2005). However, some gaps remained in the case of DE-Obe (soil temperatures), DE-Gri (NEE) and DE-Kli (soil temperatures and NEE). These remaining gaps were filled using moving averages as described above.

In the case of DE-Kli it should be noted that longer data gaps exists in 2008/2009 and 2013. A pragmatic approach was
chosen in order to arrive at complete yearly budgets. Daily values of the next following year with the same crop were used for gap-filling. Furthermore, bare soil or conditions with almost no soil cover, which occur between harvest and sowing of the next winter crop are underrepresented in the data, as the measurement equipment must be at least partially removed and reinstalled in most cases for harvest and sowing.

Soil water content measurements used in the subsequent analysis were not gap-filled. In the case of DE-Obe, soil water
content measurements were not available during the extremely dry summer 2018.





### 3.5 Principal component analysis

Principal component analysis (PCA) was used for analysing the possibly changing dependency of the different ET-estimates on meteorological variables and soil water content with changing season. Here, the 'precomp' function of package 'stats' (version 3.6.1) of R (R Core Team 2019) was used. Different combinations of variables were tested including net radiation (Rn) , global radiation (RG), vapour pressure deficit (VPD), air temperature (Tair), wind speed (WS), soil water content (SWC), energy balance closure gap (EB_gap) and evapotranspiration (ET) out of which RG, VPD, SWC and ET produced largest explained variance of the data by the first two principal components. We used scaled data for PCA because chosen variables have different dimensions. Furthermore, only daily values not subject to gap-filling were used for this analysis. Data were binned according to months prior performing PCA. PCA was then applied to all data of months December, January, February, to all data of months March, April, May, to all data of months June, July August and to all data of months September, October, November. This was done for every ET_estimate (ET_uncorr, ET_corr, ET_residual).

Please note that in the case of DE-Obe (coniferous site at a higher altitude than the other three sites) no SWC-measurements were available during the warm and extremely dry summer 2018.

## 4. Results

### 4.1 Energy balance closure

Energy balance closure of all four sites were investigated including and excluding heat storage changes due to changes in temperature ($J_H$), changes in humidity ($J_{LE}$), changes in biomass temperatures ($J_{veg,}$ only determined for DE-Tha and DE-Obe), heat storages changes between the heat flux plate and the soil surface ($J_G$) and heat storage changes due to photosynthesis and respiration ($J_C$). This analysis was based on coefficient of determination ($R^2$), simple linear regression ((H+LE) = a + b(Rn-G)) and the energy balance ratio (Eq. 6), which was obtained as follows for each year as well as over all years.

$$EBR = \frac{\sum(H+LE)}{\sum AE} ,$$
(6)

where EBR denotes energy balance ratio (dimensionless), H sensible heat flux, LE latent heat flux and AE available energy. H, LE and AE are given in W m$^{-2}$. If heat storage changes are considered, AE equals net radiation (Rn) minus ground heat flux (G) minus the sum of the respective heat storage changes. If heat storages changes are not considered, AE equals Rn – G.

Over the whole period of 11 years as well as for each year, energy balance closure expressed as energy balance ratio (Eq. 6) was very similar whether including or excluding heat storage changes (Table 3). All other investigated statistical measures (not shown) did also support this. The coefficient of determination $R^2$ changed by up to $\pm\ 0.02$ if storage terms were included. Most changes were smaller than this upper bound. For all sites the offset a of the simple linear regression only slightly changed (maximum absolute change < 1.5 W m$^{-2}$). Slopes of the simple linear regressions were also only slightly altered and most changes indicated an improvement (maximum absolute change ≤ 0.03).





Therefore, we neglected these minor terms in the subsequent analysis of evapotranspiration and latent heat fluxes, respectively, as they are of minor importance.


Table 3: Annual energy balance ratios (Eq. 6) and energy balance ratio over all 11 inspected years excluding (and including) heat storage changes. Values considering heat storage changes are given in brackets.

|  | 2008 | 2009 | 2010 | 2011 | 2012 | 2013 | 2014 | 2015 | 2016 | 2017 | 2018 | All years |
|---|---|---|---|---|---|---|---|---|---|---|---|---|
| DE-Tha | 0.64 | 0.59 | 0.67 | 0.59 | 0.66 | 0.72 | 0.76 | 0.72 | 0.73 | 0.76 | 0.78 | 0.69 |
|  | (0.64) | (0.59) | (0.68) | (0.60) | (0.66) | (0.72) | (0.77) | (0.73) | (0.74) | (0.76) | (0.78) | (0.70) |
| DE-Gri | 0.65 | 0.65 | 0.62 | 0.64 | 0.61 | 0.61 | 0.67 | 0.64 | 0.64 | 0.66 | 0.65 | 0.64 |
|  | (0.65) | (0.65) | (0.62) | (0.64) | (0.62) | (0.61) | (0.67) | (0.64) | (0.64) | (0.66) | (0.65) | (0.64) |
| DE-Kli | 0.77 | 0.64 | 0.79 | 0.84 | 0.70 | 0.91 | 0.80 | 0.76) | 0.63 | 0.47 | 0.62 | 0.71 |
|  | (0.77) | (0.65) | (0.79) | (0.85) | (0.70) | 0.92 | (0.80) | (0.77) | (0.63) | (0.47) | (0.62) | (0.72) |
| DE-Obe | 0.87 | 0.95 | 0.78 | 0.74 | 0.82 | 0.79 | 0.72 | 0.72 | 0.66 | 0.72 | 0.76 | 0.77 |
|  | (0.87) | (0.95) | (0.78) | (0.75) | (0.83) | (0.80) | (0.73) | (0.72) | (0.67) | (0.72) | (0.76) | (0.78) |

In 2017, the energy balance ratio was unusually low at DE-Kli. This year was rather a normal year concerning wetness (Fig.
2) but Bowen's ratio was quite large and indicates problems with the measurement of LE. Therefore, data of this year should be interpreted carefully.

**4.2. Results on annual scale**

Figure 3 shows yearly sums of ET of all four sites regarding different estimates of ET complemented by yearly sums of (corrected) precipitation.






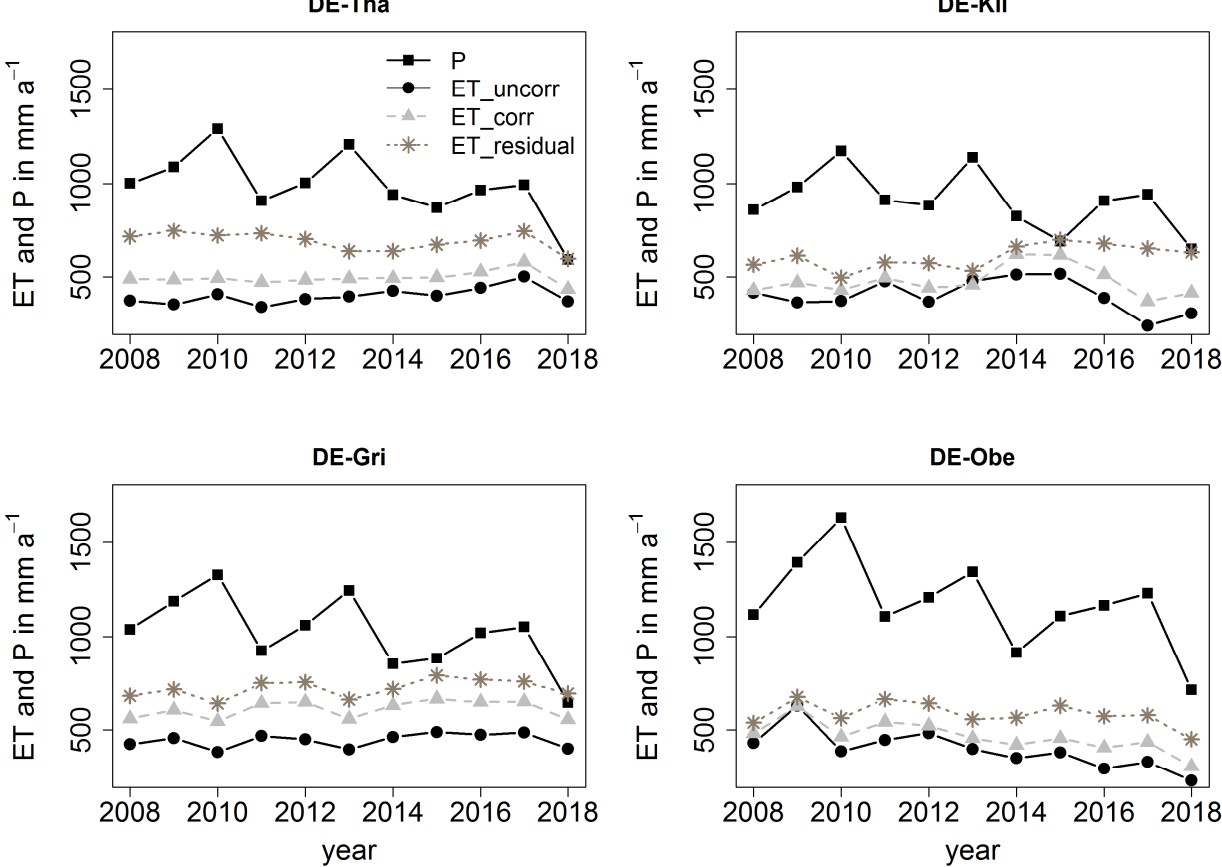

**Figure 3: Annual sums of ET-estimates and precipitation of all investigated sites for the period 2008 - 2018. ET_uncorr usually constituted the lowest estimates of ET of all sites while ET_residual constituted the largest estimates with ET_corr in between.**

Differences between the estimates were calculated as ET_corr minus ET_uncorr and ET_residual minus ET_uncorr (Table

4). Obtained differences changed from year to year and varied between -21 mm $a^{-1}$ and 200 mm $a^{-1}$ in the case of ET_corr and between 47 mm $a^{-1}$ and 413 mm $a^{-1}$ in the case of ET_residual.

Yearly differences between ET_uncorr and ET_corr and ET_residual, respectively, were closely related to energy balance closure gap of the respective year (c.f. Table 3 and Table 4). Differences plotted against yearly EBR showed almost a linear relationship (not shown) as could be expected according to the applied methods. This held for all sites except for DE-Gri

(grassland site), a site with a comparatively invariant EBR (c.f. Table 3).

In 2018 (drought year), ET_residual was larger than the corresponding total precipitation for two out of four sites (DE-Tha, DE-Gri, Fig. 3 and Table 4). In the case of DE-Kli, ET_residual was almost equal to total precipitation in 2018. ET_residual remained considerably smaller than precipitation only in the case of DE-Obe, which received more precipitation than the other three sites in 2018 (Table 4). This indicated that the size of ET_residual can be questioned with respect to the yearly

sum of precipitation during years of intense droughts.





ET_uncorr was larger than ET_corr for two site years. Those were the years 2013 at DE-Kli and 2009 at DE-Obe. In 2013, a comparatively high amount of data must be gap-filled in the case of DE-Kli. Data of other years when the same crop was grown were used for gap-filling. The amount of gap-filled data did not explain why ET_uncorr is very similar to ET_corr at DE-Obe in 2009. However, August 2009 was the month with most gaps at DE-Obe in 2009, a month with comparatively 325 large ET. We can, therefore, not exclude that this result is due to gap-filling.

Table 4:Annual sums of ET_uncorr, ET_corr, ET_residuals and precipitation (P) in mm a$^{-1}$ for all sites. Grey rows display the differences of ET_corr and ET_residuals in relation to ET_uncorr.

| DE-Tha | 2008 | 2009 | 2010 | 2011 | 2012 | 2013 | 2014 | 2015 | 2016 | 2017 | 2018 | cumulative |
|---|---|---|---|---|---|---|---|---|---|---|---|---|
| P | 1001 | 1087 | 1290 | 908 | 1003 | 1207 | 941 | 871 | 965 | 994 | 592 | 10860 |
| ET_uncorr | 373 | 352 | 407 | 338 | 382 | 394 | 426 | 399 | 441 | 501 | 368 | 4382 |
| ET_corr | 489 | 485 | 494 | 472 | 484 | 491 | 494 | 497 | 527 | 580 | 434 | 5446 |
| ET_residual | 717 | 746 | 722 | 733 | 702 | 637 | 638 | 672 | 695 | 746 | 597 | 7605 |
| ET_corr - ET_uncorr | 116 | 132 | 87 | 133 | 103 | 96 | 68 | 98 | 86 | 79 | 66 | 1064 |
| ET_residual - ET_uncorr | 344 | 394 | 315 | 395 | 320 | 243 | 213 | 274 | 253 | 244 | 228 | 3223 |
| DE-Gri | 2008 | 2009 | 2010 | 2011 | 2012 | 2013 | 2014 | 2015 | 2016 | 2017 | 2018 | cumulative |
| P | 1039 | 1187 | 1328 | 924 | 1061 | 1245 | 854 | 884 | 1019 | 1051 | 647 | 11238 |
| ET_uncorr | 423 | 456 | 381 | 468 | 449 | 395 | 463 | 489 | 474 | 486 | 399 | 4883 |
| ET_corr | 560 | 606 | 545 | 643 | 649 | 559 | 632 | 666 | 649 | 651 | 555 | 6717 |
| ET_residual | 682 | 717 | 641 | 751 | 754 | 663 | 719 | 791 | 769 | 758 | 694 | 7939 |
| ET_corr - ET_uncorr | 137 | 150 | 164 | 176 | 200 | 164 | 169 | 178 | 175 | 165 | 156 | 1834 |
| ET_residual - ET_uncorr | 259 | 261 | 260 | 283 | 305 | 268 | 256 | 302 | 295 | 272 | 295 | 3056 |
| DE-Kli | 2008 | 2009 | 2010 | 2011 | 2012 | 2013 | 2014 | 2015 | 2016 | 2017 | 2018 | cumulative |
| P | 859 | 982 | 1173 | 912 | 884 | 1139 | 825 | 691 | 908 | 942 | 650 | 9963 |
| ET_uncorr | 416 | 363 | 370 | 477 | 366 | 479 | 513 | 515 | 386 | 238 | 304 | 4428 |
| ET_corr | 428 | 470 | 430 | 495 | 442 | 457 | 621 | 618 | 513 | 369 | 415 | 5258 |
| ET_residual | 566 | 614 | 497 | 577 | 574 | 532 | 661 | 698 | 677 | 651 | 632 | 6680 |
| ET_corr - ET_uncorr | 13 | 106 | 59 | 19 | 76 | -21 | 108 | 102 | 127 | 131 | 111 | 830 |
| ET_residual - ET_uncorr | 150 | 250 | 126 | 101 | 208 | 54 | 148 | 183 | 291 | 413 | 328 | 2252 |
| DE-Obe | 2008 | 2009 | 2010 | 2011 | 2012 | 2013 | 2014 | 2015 | 2016 | 2017 | 2018 | cumulative |
| P | 1117 | 1394 | 1634 | 1107 | 1208 | 1343 | 914 | 1108 | 1165 | 1230 | 714 | 12933 |
| ET_uncorr | 429 | 630 | 385 | 446 | 483 | 397 | 348 | 380 | 292 | 329 | 228 | 4345 |
| ET_corr | 481 | 627 | 463 | 543 | 523 | 455 | 420 | 456 | 405 | 435 | 304 | 5112 |
| ET_residual | 538 | 676 | 563 | 665 | 641 | 556 | 566 | 629 | 574 | 578 | 451 | 6437 |
| ET_corr - ET_uncorr | 52 | -3 | 79 | 97 | 40 | 58 | 72 | 76 | 113 | 106 | 77 | 767 |
| ET_residual - ET_uncorr | 109 | 47 | 179 | 219 | 158 | 159 | 218 | 250 | 282 | 249 | 223 | 2092 |





Variability of yearly sums of ET_uncorr was largest at all sites as indicated by the variation coefficient (DE-Tha: 11.4%, DE-Gri: 8.6%, DE-Kli: 22.8%, DE-Obe: 26.8%) and decreased from ET_uncorr to ET_corr (7.2%, 7.6%, 16.7%, 17.8%) and ET_residual (7.2%, 6.6%, 10.5%, 11.1%). The old spruce site (DE-Tha) showed a lower variability in ET than the younger spruce site (DE-Obe). DE-Tha and DE-Obe are characterized by tall vegetation. DE-Gri and DE-Kli are characterized by low vegetation. Here, the strongly managed crop site DE-Kli showed larger variation than the less managed grassland site DE-

Gri.

Largest yearly precipitation (P) does not necessarily coincide with largest ET indicating the irregular seasonal distribution of P. Furthermore, other driving factors like available energy are not addressed in Fig. 3, but are important. The latter is an important point as ET of the selected study region is rather limited by energy than by P (c.f. Fig.3). Figure 4 addresses this issue and shows yearly evaporative fractions (LE/Rn), ET/P as well as (Rn/L)/P. Commonly, the ratio ET/P is smaller than

the ratio ET/Rn, indicating that a larger fraction of available energy is used for ET than of received precipitation. However, the ratio ET/P was larger than the ratio ET/Rn when (Rn/L)/P was larger than one. These were years where more energy was supplied by Rn than would have been needed to evaporate the yearly sum of P. The ratios of ET/P and ET/Rn of these years suggest that limitation of ET of these years by P was more prominent than limitation of ET by Rn (rather P-limited than energy limited). Figure 4 further shows that the partition of available energy to sensible heat and latent heat would change

with used estimate of ET. The used fraction of Rn for ET generally increases from ET_uncorr to ET_corr and ET_residual for all sites. Exceptions are the two site years discussed above. However, ET_residual always used the largest fraction of Rn. In the case of DE-Gri (grassland site, soil with large water holding capacity) ET_residual almost totally consumed Rn very often.





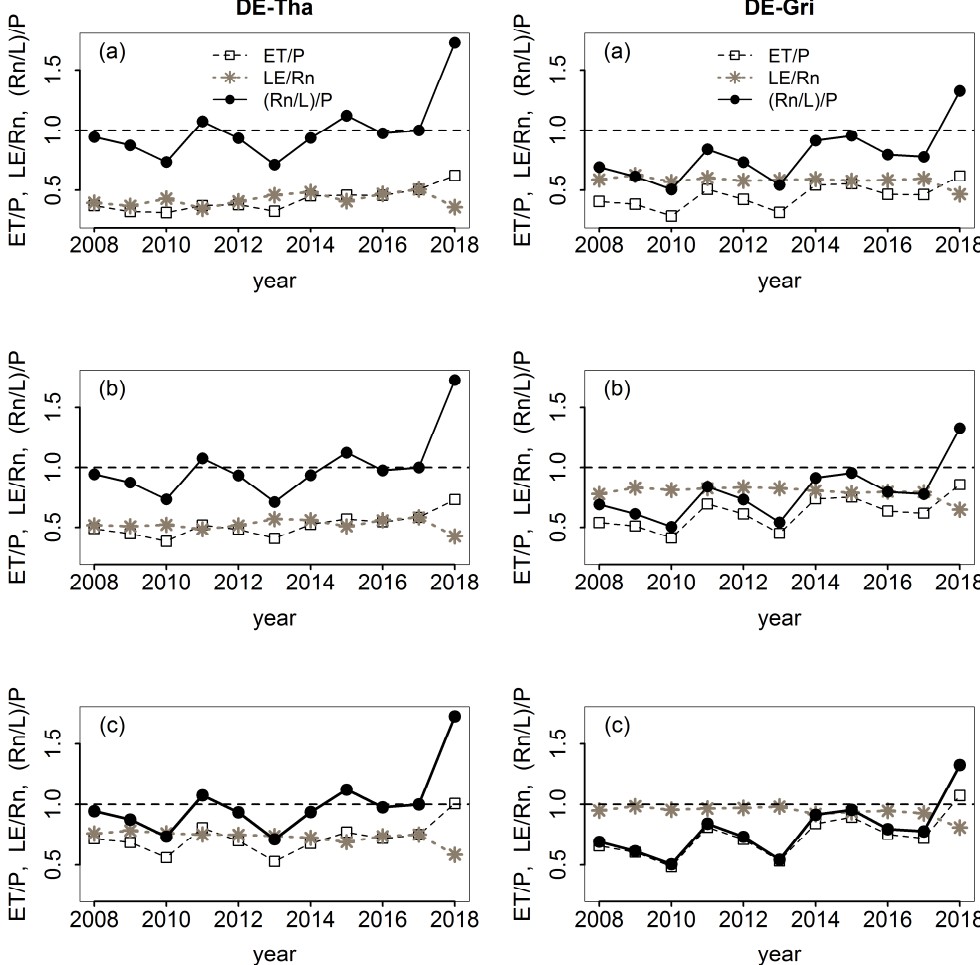

**Figure 4: Yearly ratios of ET/P, ET/Rn and (Rn/L)/P using ET_uncorr, ET_corr und ET_residual. Left graph refers to DE-Tha and right graph refers to DE-Gri. Subplot (a) refers to ET_uncorr, subplot (b) to ET_corr, and subplot (c) to ET_residual. Rn refers to net radiation and L latent heat of vaporization.**

All different estimates of ET showed no clear relationship when inspected as a function of Rn and P. However, there is a slight tendency to smaller ET when either Rn is small but P is large or P is small and Rn is large. DE-Kli with its crop rotation management did not show this tendency. These results confirmed that inter-annual variation of these two variables is important.

## 4.3 Results at seasonal scale

### 4.3.1 Intra-annual variation

Figure 5 shows the average yearly courses of all three ET estimates over the investigated period for all four sites. In order to display complete yearly courses gap-filled data were used.





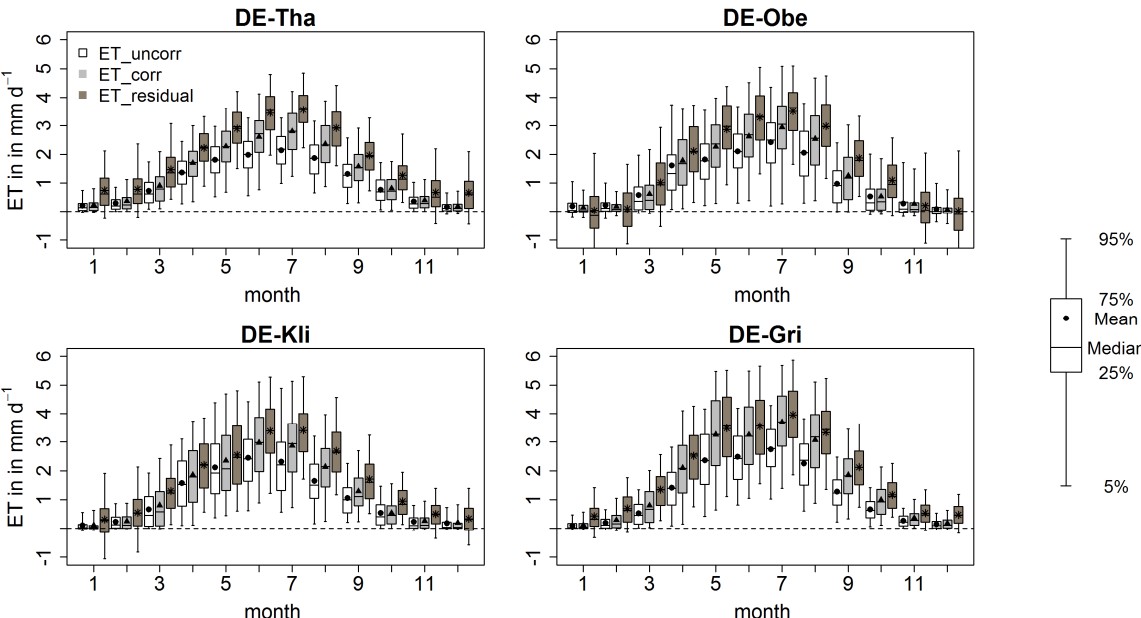

**Figure 5: Monthly boxplots of daily values of ET_uncorr, ET_corr and ET_residual based on gap-filled data.**

All estimates share a similar annual course with largest values in summer and lowest values in winter and tend to peak in June/July. ET_residual generally show largest values, whereas ET_uncorr show smallest values with ET_corr in between. The interquartile range is considerably smaller in colder month for ET_uncorr and ET_corr than for ET_residual.

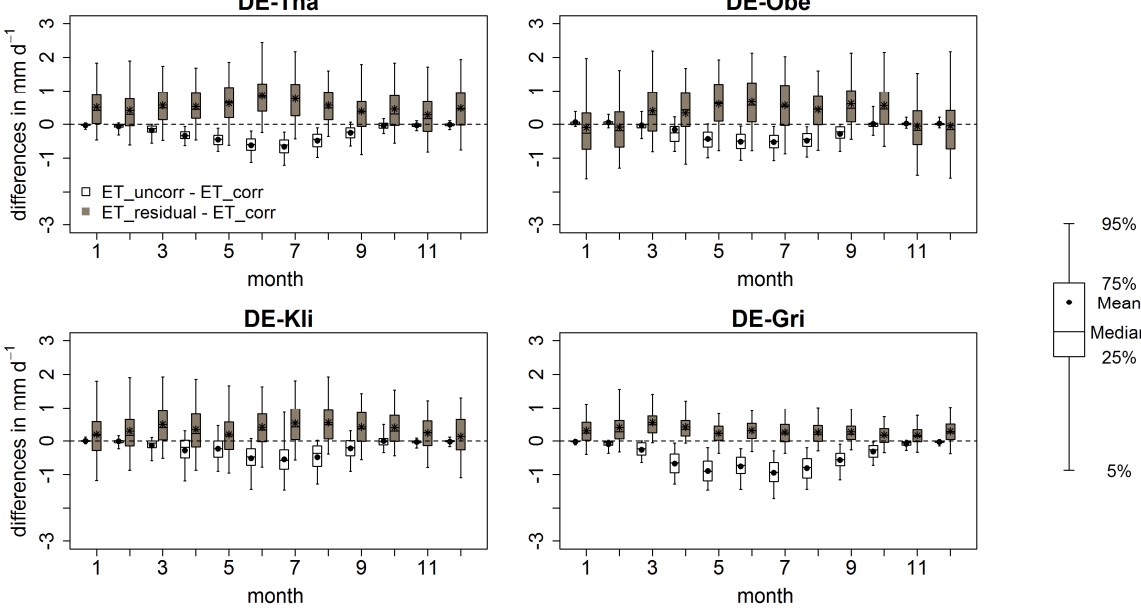

**Figure 6: Monthly boxplots of daily values of ET_uncorr minus, ET_corr and ET_residual minus ET_corr based on gap-filled data.**





The differences (Fig. 6) between ET_uncorr and ET_corr show a well marked yearly course with larger values during
summer than during winter. Differences between ET_uncorr and ET_corr are primarily negative over the whole year,
indicating that ET_uncorr underestimates ET in relation to ET_corr. In contrast to this, the differences between between
ET_residual and ET_corr show no clear yearly course and their interquartile range is comparatively invariant over the year.
This indicates that ET_residual overestimates ET in relation to ET_corr in relative terms more during the colder months than
during the warmer months.

Differences between ET_residual and ET_corr are more often negative in the case of DE-Kli (agricultural site) and
especially in the case of DE-Obe (coniferous site) during winter months.

**Figure 7: Linear correlation between ET_corr and ET_uncorr (black open circles) and ET_corr and ET_residual (red asterisks).
Only daily values were used, which were not subject to gap filling. Upper panel shows results for DE-Tha (old coniferous forest)**
**and lower panel shows results for DE-Gri (grassland site). For equations of simple linear regressions please refer to Table 5.**





Although all three estimates of ET share a similar mean yearly course, differences indicate larger deviations for the colder months. This is confirmed when inspecting correlation between the different estimates for different months of the year (Fig. 7 and Table 5). Only daily values were used, which were not subject to gap filling.

All sites showed least agreement between ET_corr and ET_residual for December, January and February. During these months ET_corr and ET_residual are totally uncorrelated. Best agreement was found for the months March, April and May but correlation is still very weak except for DE-Gri (grassland site). The offset of the linear regression line is largest for sites during June, July and August. The values were 1.82 mm d$^{-1}$, 0.76 mm d$^{-1}$, 1.35 mm d$^{-1}$, 1.66 mm d$^{-1}$ in the case of DE-Tha (old coniferous site), DE-Gri (grassland site), DE-Kli (crop site) and DE-Obe (coniferous site).

DE-Gri, in typical years with little water stress, showed deviating results in case of the comparison of ET_corr to 390 ET_residual (Fig. 6). As for the other sites, results of correlation analysis of months December, January and February indicated that ET_corr and ET_residual are uncorrelated. However, R² considerably improved for the other inspected months, which indicated that the relative course agreed reasonably well but absolute overestimation by ET_residual persisted.

For all sites ET_uncorr and ET_corr correlate well, as ET_corr is based on ET_uncorr and they are therefore not independent 395 of each other.


Table 5: Results of linear regressions, R² denotes coefficient of determination. Slope a and offset b of linear regression line (y = a + bx) is given too. Numbers used for linear regression varied between 812 and 935. Please note that only days were used which were not subject to gap filling.

| | Dec Jan Feb | Mar Apr May | Jun Jul Aug | Sep Oct Nov |
|---|---|---|---|---|
| DE-Tha | | | | |
| ET_corr vs. ET_uncorr | | | | |
| R² | 0.94 | 0.97 | 0.94 | 0.94 |
| Slope b | 0.79 | 0.8 | 0.77 | 0.84 |
| Offset a in mm d$^{-1}$ | 0.02 | -0.01 | 0.00 | 0.04 |
| ET_corr vs. ET_residual | | | | |
| R² | 0.07 | 0.54 | 0.42 | 0.33 |
| Slope b | 0.56 | 0.79 | 0.59 | 0.68 |
| Offset a in mm d$^{-1}$ | 0.58 | 0.91 | 1.82 | 0.69 |
| DE-Gri | | | | |
| ET_corr vs. ET_uncorr | | | | |
| R² | 0.94 | 0.99 | 0.96 | 0.97 |
| Slope b | 0.72 | 0.73 | 0.74 | 0.68 |
| Offset a in mm d$^{-1}$ | 0.01 | -0.06 | 0.02 | 0.01 |
| ET_corr vs. ET_residual | | | | |
| R² | 0.13 | 0.92 | 0.90 | 0.85 |
| Slope b | 0.69 | 0.87 | 0.86 | 0.93 |
| Offset a in mm d$^{-1}$ | 0.38 | 0.67 | 0.76 | 0.29 |
| DE-Kli | | | | |
| ET_corr vs. ET_uncorr | | | | |
| R² | 0.94 | 0.93 | 0.85 | 0.83 |
| Slope b | 0.78 | 0.84 | 0.84 | 0.81 |
| Offset a in mm d$^{-1}$ | 0.03 | 0.04 | -0.07 | 0.06 |
| ET_corr vs. ET_residual | | | | |
| R² | 0.02 | 0.64 | 0.68 | 0.46 |
| Slope b | 0.24 | 0.78 | 0.69 | 0.80 |
| Offset a in mm d$^{-1}$ | 0.32 | 0.78 | 1.35 | 0.52 |
| DE-Obe | | | | |
| ET_corr vs. ET_uncorr | | | | |
| R² | 0.91 | 0.78 | 0.95 | 0.91 |
| Slope b | 1.2 | 0.92 | 0.84 | 0.82 |
| Offset a in mm d$^{-1}$ | 0.02 | -0.07 | -0.09 | 0.03 |
| ET_corr vs. ET_residual | | | | |
| R² | 0.06 | 0.55 | 0.48 | 0.37 |
| Slope b | 0.49 | 0.80 | 0.60 | 0.76 |
| Offset a in mm d$^{-1}$ | -0.02 | 0.74 | 1.66 | 0.60 |





### 4.3.2 Dependency on meteorological variables – principal component analysis

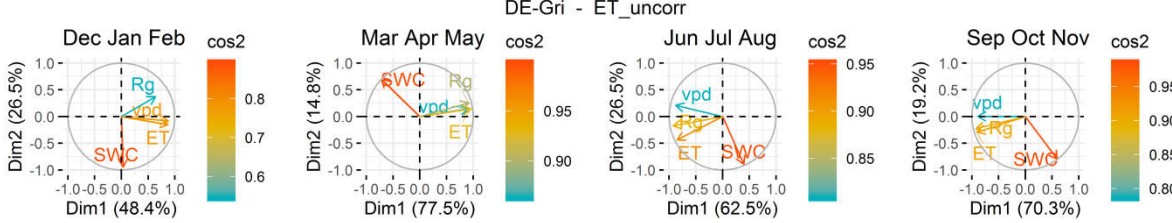

**Fig. 8: Results of PCA for data grouped according to month. Only days not subject to gap-filling were used. Coordinates of arrow heads represent the correlation of the respective variable with principal component 1 (Dim1) and 2 (Dim2), respectively. The cos2-value is the squared value of the coordinates and ranks the variables according to their representation in the respective principal component. Number of used data points: Dec, Jan, Feb.: 663; Mar, Apr, May: 677; Jun Jul Aug: 716; Sep Oct, Nov: 717.**

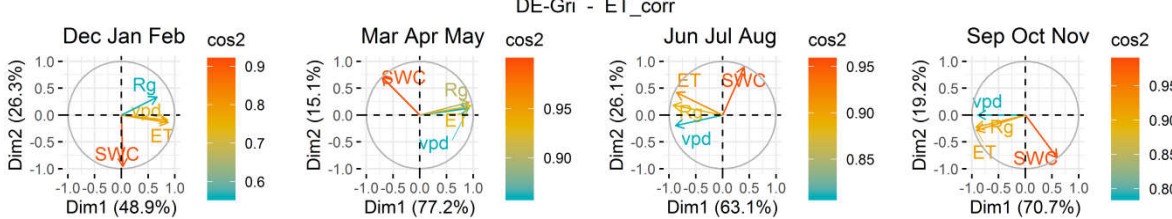

**Fig. 9: Same as Fig. 8 but for ET_corr.**

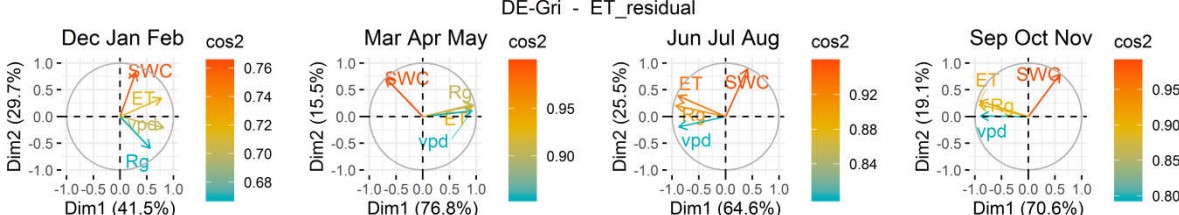

**Fig. 10: Same as Fig. 8 but for ET_residual.**

Principal component analysis (PCA) was performed using the variables global radiation, vapour pressure deficit, soil water content and respective ET estimates. Figure 8, 9 and 10 shows the factor map of DE-Gri (grassland site) for each ET-estimate and grouped months.

For the chosen combination of variables (RG, VPD, SWC, ET) the first two principal components (PC) explained most of the variance of the data for the months March, April and May (between 77% and 92%) for all four sites. Least variance was explained in the months December, January and February (between 63% and 71%). This ranking did not depend on the chosen ET estimate or on inspected site. Explained variance by the first two PCs was always lowest for ET_residual for all sites in December, January and February.

The large discrepancy between ET_corr and ET_residual as well as ET_corr and ET_residual suggests a possibly deviating dependency on meteorological variables. Therefore, we analysed which variables contributed most to the first and second principal component (PC1 and PC2, respectively). However, no consistent differences could be detected. RG mostly





contributed most to PC1, but contributions of VPD and ET were often of similar magnitude and sometimes slightly larger. RG contributed most to PC1 for all sites in the warmer months (June – November). Soil water content always dominated PC2. Table B1, B2, B3, B4 (Appendix B) give an overview over all contributions of all variables to all PCs.

## 5. Discussion

### 5.1 Yearly values

Results of annual evapotranspiration showed largest values for ET_residual and lowest values for ET_uncorr whereas ET_corr was in between. This agrees with the study of Barr et al. (2012). They investigated ten years data of different sites all located in the White Gull Creek watershed (Canada) and obtained best agreement between estimated and measured outflow when using estimates of ET corrected for the energy balance closure gap. Using ET_residual instead of ET_corr yielded a slightly negative outflow indicating an overestimation of ET. Their obtained mean annual difference between ET_uncorr and ET_residual was 173 mm a$^{-1}$ and corresponds to the lower bound of mean annual differences obtained in this study (c.f. Table 4).

Gebler et al. (2015) compared lysimeter based ET data to ET corrected for energy balance closure and to ET_residual. Best agreement was found between ET corrected for an energy balance closure gap and lysimeter data whereas ET_residual overestimated lysimeter data by 15% on average. Assuming that our estimate ET_corr would also be closest to the true value of ET, overestimation of ET by ET_residual would be even larger and between 20 and 40 % based on mean annual values.

Unfortunately we lack lysimeter data or runoff data for an independent evaluation of our different ET-estimates. However, we are able to review results with regard to a possible water and energy limitation. All annual estimates of ET_residual are plausible with respect to received precipitation (except two sites in 2018) and radiation. For a commonly well watered grassland (DE-Gri) yearly values of ET_residual represents the maximum attainable ET (Fig. 11) constituting an upper estimate of ET with regard to supplied Rn. This is not the case for the other sites where ET_residual do not follow the 1:1 line of the Budyko-curve (Fig. 11). We hypothesize that different water management strategies of the respective landuse (grassland vs forest), management (DE-Kli) as well as gap-filling (particularly in the case of DE-Kli and DE-Obe) might be an issue for the differing results of the differing sites.



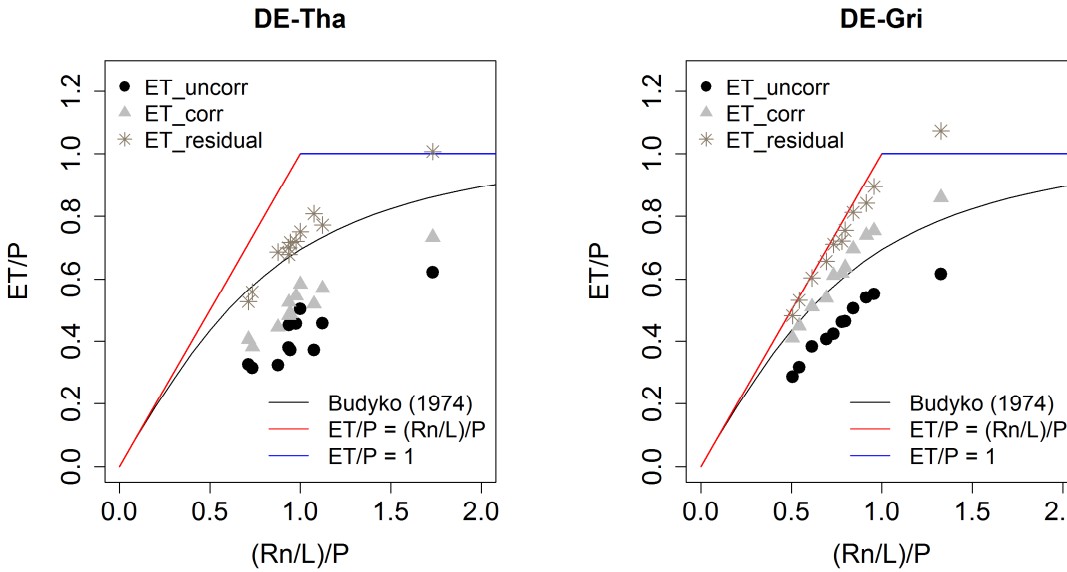


**Figure 11: Annual sums of ET based on different methods assessed in a framework following Budyko (1974). Right graph refers to DE-Tha (old spruce site) and left graph refers to DE-Gri (grassland).**

Again, the year 2018 (drought year) constitutes a special case, where for two sites (DE-Tha and DE-Gri) the yearly sum of ET_residual is not supported by the received precipitation but would have been possible with respect to supplied energy by 450  net radiation (Fig. 4). This indicates that yearly P might be used as a plausibility check in the case of ET_residual whether these large estimates are possible at all (assuming that all available water of the respective year for ET is supplied by P). However, single dry years can profit from proceeding wetter years and plants might be able tap water from deeper layers. For the investigated sites, longterm P was always larger than ET (Fig. 12).

The large differences between the estimates indicated that different estimates would drastically change obtained water 455  balance as already demonstrated by Barr et al. (2012) and Gebler et al. (2015) also on an annual basis. These large differences produced inter-annual variations, which considerably changed with different ET estimates (Fig. 3). This should be kept in mind when differences in ET between sites are assessed, as relation between sites with regard to ET might considerably change with different estimates. Gaps in the energy balance closure are site specific resulting in site specific underestimations of ET. Therefore, uncorrected ET indicate not only implausible low numbers regarding water budget but 460  also an unrealistic large land use variability of ET. This underlines the need of correction of ET for energy balance closure gaps to avoid spurious land use dependencies.

Results further showed that on annual scale there is not necessarily a strong relationship between annual P and annual Rn. On one hand, this highlighted the importance of intra-annual variation of P and Rn. On the other hand, we have to consider management as (e.g.) management can change from year to year according to the crop grown at DE-Kli (crop site).





Furthermore, influences of gap-filling could not be excluded (especially in the case of DE-Kli and DE-Obe) but are unavoidable for obtaining annual sums.

**Figure 12: Cumulative sums of ET and P for all sites and the whole investigated period (years 2008 – 2018)**

**5.2 Discussion intra-annual variation**

Differences between ET_uncorr and ET_corr are primarily negative over the whole year, indicating that ET_uncorr underestimates ET in relation to ET_corr. Largest differences between ET_uncorr and ET_corr were found for DE-Gri (grassland site), which was the site with lowest energy balance ratio (EBR) when averaged over all years, followed by DE-THA (old coniferous site), which showed second lowest EBR (c.f. section 4.1 Energy balance closure). The overestimation

of ET_residual in relation to ET_corr is in accordance with other studies (e.g. Wohlfahrt et al. 2010; Barr et al. 2012; Gebler et al. 2015, Mauder et al. 2018).



In relative terms ET_residual is more often negative in the colder months in the case of DE-Kli (crop site) and DE-Obe (coniferous site) and large negative values could frequently occur. Excluding all days with missing half-hourly data, i.e. using only days with measured 48 half-hourly values of LE, H, Rn and G did not change this picture, therefore we assume
that gap-filling is of minor importance here. We further tested whether this result could be related to snow coverage as these two sites (DE-Kli and DE-Obe) are situated at higher altitudes than DE-Gri (grassland site) and DE-Tha (old coniferous forest). Excluding days with snow, based on available information, reduced times with negative ET_residual at DE-Kli but not necessarily at DE-Obe. Therefore, we analysed the corresponding values of Rn, H and G in the case of DE-Obe. G was always close to zero for negative values of ET_residual. Large negative values of ET_residual occurred when either both Rn
and H were small but positive and H was larger than Rn or Rn was comparatively large but negative and H close to zero. It should be noted that in both cases the energy supply by radiation for a possibly positive ET was rather limited and other energy sources than radiation must have sustained a possibly positive ET, e.g. energy supply by heat storage changes. However, obtained cumulative heat storage changes explained only a minor portion of positive values of ET_corr when Rn as well ET_residual were negative. We hypothesize that this is due to uncertainties in the calculation of the heat storage
changes and other energy balance components. However, daily values of heat storages changes are rather small and often negligible as positive contributions over the day are cancelled out by negative contributions during nighttime. Sub-daily time scales are needed to investigate the importance of heat storage changes as a possible energy source for evaporation during winter.

When inspecting differences between ET_residual and ET_corr, it became apparent that ET_corr is more overestimated
during winter months in relative terms than during summer months. Additionally, differences of both estimates can be of opposite sign. We hypothesize that this is mainly due to measurement uncertainties as involved fluxes are small during winter and little absolute changes can result in large relative errors (c.f. McNeil and Shuttleworth 1975, Barr et al. 2012). Therefore, ET_residual should be used very cautious during winter months as it has been already noted by Amiro et al. (2009).

This result was confirmed by simple linear regressions performed for different seasons of the year, which indicated uncorrelated ET estimates of ET_corr in relation to ET_residual. However, results of DE-Gri (a site with little water stress in typical years) showed reasonably coefficients of determination ($R^2 > 0.85$) except for the months December, January and February. We assume that this result is obtained due to the fact that this site very rarely experiences low soil moisture content and also due to the different water use strategies of grass compared to trees.

We note, that DE-Tha (old coniferous forest) tend to show lower maximal values of ET than DE-Obe (coniferous forest) although DE-Tha is situated at an lower altitude than DE-Obe, which calls for reviewing the general assumption of decreasing ET with increasing altitude (e.g. Baumgartner et al. 1982/1983; Goulden et al. 2012). We hypothesize, that rainfall interception is important here. DE-Obe commonly receives more rainfall than DE-Tha. This means the canopy is more often wetted at DE-Obe than at DE-Tha. Therefore, rainfall interception occurs more often at DE-Obe. When
interception evaporation takes places, the surface of the canopy cools down and reverses the vertical temperature gradient





(surface of canopy and adjacent air layers is cooler than air layers above) which facilitates a sensible heat flux now directed towards the earth surface (negative sensible heat flux). This negative sensible heat flux serves as an additional energy input for further interception evaporation. Of course this is also true for DE-Tha. The difference might be larger wind speeds at DE-Obe compared to DE-Tha. This can facilitates higher interception evaporation despite a low VPD. Additionally, DE-Obe

is more often cloudy compared to DE-Tha and more cloud-water is intercepted at this site and can accordingly enhance interception evaporation. However, rainfall interception cannot be resolved using daily values as it typically is a process which takes place at sub-daily timescale. The issue of rainfall interception will be more detailed addressed in subsequent research.

## 5.3 Discussion PCA-results

The dependency of the different ET estimates on driving meteorological variables were analysed using principal component analysis (PCA) using the variables RG, VPD, SWC and ET.

Results showed that commonly RG contributed most to the first principal component, but contributions of VPD and ET itself were often of similar magnitude. These variables dominate PC1 as they are closely correlated.

In contrast to this, SWC dominates the second principal component (PC2). The difference in explained variance between

PC1 and PC2 is most pronounced for March, April May. Explained variance by PC1 decreases from March, April, May to June, July, August whereas the variance explained by PC2 increases. This can be explained by the fact that soil moisture is commonly plenty available during March, April, May. Therefore, ET of March April, May depends more on variables contributing to PC1 (e.g. RG). Soil moisture availability often decreases over the summer month whereas the inputted energy by radiation is comparatively large. Therefore variance explained by PC2 increases from March, April, May to June, July

August, indicating that the importance of SWC increased.

PC1 and PC2 explained least variance of the data for December, January, February, which is a consistent result for all sites. One reason might be measurement uncertainty as ET values are comparatively small during winter months and therefore the relative uncertainty increases. Explained variance by PC1 and PC2 was always lowest in the case of ET_residual during December, January, February. This was a consistent result but differences in explained variance by PC1 and PC2 compared

to ET_uncorr and ET_corr were rather small. A cautious interpretation would be that ET_residual is somewhat less dependent on the contributing variables to PC1 and PC2 as the errors of all other components of the energy balance pile up in ET_residual (McNeil and Shuttleworth 1975; Barr et al. 2012).

## 6. Summary and conclusion

Three different estimates for ET were compared to each other for four sites differing in land-use and also in altitude at a

daily time-scale for 2008 – 2018 (11 years) on a daily basis. These three ET estimates were, ET_residual based on the residual of the energy balance, ET_corr (corrected for the energy balance closure gap) and ET_uncorr (not corrected for the energy balance closure gap). ET_residual delivered largest values. On average, it was 196 mm a$^{-1}$ (111 mm a$^{-1}$, 129 mm a$^{-}$



[1],121 mm a[-1]) larger than ET_corr at DE-Tha (DE-Gri, DE-Kli, DE-Obe). ET_uncorr showed lowest values and was 97 mm a[-1] (167 mm a[-1], 75 mm a[-1],70 mm a[-1]) lower than ET_corr on average at DE-Tha (DE-Gri, DE-Kli, DE-Obe). The

differences between the different estimates were site-specific and were closely related to the respective energy balance closure gap.

ET_uncorr is affected by undetected latent heat fluxes. It can represent a reasonable lower estimate of ET but underestimates ET due to the inherent energy balance closure gap. ET_residual is represents a reasonable upper estimate but tends to overestimate ET from a water budget point of view. During the dry year 2018 the annual sum of ET_residual was larger than

the annual sum of P for two of the four investigated sites. Within this range ET_corr is the most reliable estimate and is recommended, especially when assessing land use dependencies of ET.

ET_uncorr was slightly larger on a yearly basis than ET_corr for two site years. This was most likely an issue of gap-filling as the respective years were years with a large amount of gap-filled data.

During the cold season ET_residual was uncorrelated to the other two estimates of ET. We attribute this to measurement

errors of the other components of the energy balance, which are also comparatively small and therefore the relative error is comparatively large. Consequently, ET_residual should be used only very cautious during the cold season.

ET_residual correlated best with ET_corr and ET_uncorr at DE-GRI (grassland site). The different water use strategy compared to the coniferous sites and the moderate management compared to the crop site might be possible explanations. Therefore, ET-residual can provide reliable results for low vegetation sites that are not heavily managed and rarely exposed

to water stress during most of the year.

We also tested whether the different estimates of ET differ in their dependencies on driving variables by using principal component analysis. No large differences could be detected concerning the dependency on RG, VPD and SWC.

We noted that maximum values of ET can be larger for the coniferous site DE-Obe than the coniferous site DE-Tha despite the fact that DE-Obe is situated at an altitude 350 m higher than DE-Tha. We hypothesize that this is due to differences in

rainfall interception. However, rainfall interception is a process which takes place at sub-daily timescales. Studies at smaller time-scales than the daily scale are needed to investigate this aspect further.

**Appendices**

**Appendix A – Heat storage changes**

$J_H$ and $J_{LE}$ were calculated using air temperature and humidity changes above the canopy following Aubinet et al. (2001)

according to Eq. A1 and Eq. A2, respectively,

$$J_H \cong \frac{\rho_a \, c_p \, z \, \Delta T_{air}}{\Delta t},$$   (A1)

where $\Delta T_{air}$ denotes air temperature difference of between two consecutive time steps in K, $\rho_a$ air density in kg m[-3], cp specific heat capacity of air at constant pressure in J kg[-1] K[-1], z reference height in m and $\Delta t$ time step in s ($\Delta t$ = 1800s).

$J_{LE}$ was calculated in an analogous way,





$J_{LE} \cong \frac{L\,(T_{air})z\,\Delta\rho_v}{\Delta t}$                         (A2)

were L denotes latent heat of vaporisation in J kg$^{-1}$. L was calculated as a function of air temperature $T_{air}$. $\Delta\rho_v$. denotes difference in water vapour density between two consecutive time steps in kg m$^{-3}$. L was assumed to be constant over the whole canopy.

$J_C$ was obtained following Leuning et al. (2012) according to Eq. A3

$J_c = -\alpha_p\,NEE,$                              (A3)

where $\alpha_p$ is the photosynthetic energy conversion factor (0.469 J μmol$^{-1}$, Blanken et al. 1997) and a negative sign representing an uptake of $CO_2$. NEE is net ecosystem exchange of CO2 in μmol m$^{-2}$ s$^{-1}$. NEE was obtained via EC at the respective sites.

$J_{veg}$ was calculated for the two coniferous sites DE-Tha and DE-Obe following Thom (1975) Eq. A4.

$J_{veg} = \frac{c_{veg}\,m_{veg}\,\Delta T_{a\_fit}}{\Delta t},$                       (A4)

where, $c_{veg}$ denotes canopy specific heat capacity in J kg$^{-1}$ K$^{-1}$ whereas a value of 2958 J kg K$^{-1}$ was assumed following Thom (1975) that $c_{veg}$ is approximately 70% of the corresponding value for water. Wet biomass of vegetation in kg was calculated based on empirical findings of Sharma (1992). A fitted air temperature $T_{a\_fit}$ was used in order to reproduce bole temperature, which is damped and shifted in time in comparison to air temperature. Firstly, the air temperature was smoothed by a moving average over 5 consecutive half hourly values. Secondly, the time lag between bole temperature and air temperature was determined using cross correlation. This was done on basis of data of DE-Tha (old coniferous forests). A time lag of 3.5 h was obtained, which was accordingly applied to the smoothed air temperature. The same time-lag as for DE-Tha was assumed in the case of DE-Obe (coniferous site) as there were no bole temperature measurements available.

$J_{veg}$ was not calculated for the agricultural site DE-Kli and the grassland site DE-Gri due to missing sufficient information about fresh weight of biomass. However, former studies have shown that for sites with low vegetation as in the case of DE-Gri and DE-Kli, $J_{veg}$ is of minor importance in comparison to other heat storage changes (Jabcobs et al. 2008; Eshonkulov et al. 2019).

Heat storage change between ground heat flux plate and soil surface $J_G$ was calculated according to Moderow et al. (2009), which is a pragmatic approach using Eq. A5,

$J_G = \frac{c_s\,z_s\,\Delta T_s}{\Delta t},$                              (A5)

$c_s$ is the volumetric heat capacity of the soil in J m$^{-3}$ K$^{-1}$, $z_s$ is the depth of the heat flux plate in m, $T_s$ is soil temperature at 0.01 m depth. Values for $c_s$ were taken from Dehner et al. (2007) according to the proportion of sand, silt and clay (DE-Tha, DE-Obe, DE-Gri: 1.8 *10$^6$ J m$^{-3}$ K$^{-1}$; DE-Kli: 1.7*10$^6$ J m$^{-3}$ K$^{-1}$) assuming soil water contents of field capacity. $c_s$ was taken as constant and results in an overestimation of $J_G$ for drier conditions and an underestimation of $J_G$ for wetter conditions. $T_s$

at 0.01 m depth was obtained via linearly extrapolation using temperature from deeper depths (Table 1).





**Appendix B – Contributing variables to principal component 1 and 2**

Table B1: Contribution of each variable to principal component 1 (Dim. 1) and principal component 2 (Dim. 2). Contributions are given in %. Table refers to site DE-Tha (old coniferous forest). Shaded numbers highlights the highest contribution to the respective component.

| | DE-Tha | | | | | |
| | ET_uncorr | | ET_corr | | ET_residual | |
| Dec, Jan, Feb | Dim.1 | Dim.2 | Dim.1 | Dim.2 | Dim.1 | Dim.2 |
|---|---|---|---|---|---|---|
| Tair | 38.1 | 4.2 | 38.6 | 4.4 | 51.1 | 1.0 |
| vpd | 41.6 | 5.5 | 42.0 | 5.5 | 46.8 | 8.8 |
| SWC | 2.1 | 78.0 | 2.1 | 76.0 | 1.4 | 30.6 |
| ET | 18.2 | 12.3 | 17.2 | 14.1 | 0.7 | 59.6 |
| Mar Apr May | Dim.1 | Dim.2 | Dim.1 | Dim.2 | Dim.1 | Dim.2 |
| Tair | 31.1 | 3.1 | 31.4 | 2.7 | 32.2 | 1.8 |
| vpd | 29.7 | 2.8 | 29.8 | 2.2 | 30.6 | 1.0 |
| SWC | 14.9 | 83.2 | 14.8 | 82.2 | 15.7 | 75.9 |
| ET | 24.3 | 10.9 | 24.0 | 12.9 | 21.4 | 21.3 |
| Jun Jul Aug | Dim.1 | Dim.2 | Dim.1 | Dim.2 | Dim.1 | Dim.2 |
| Tair | 41.6 | 0.2 | 42.5 | 0.5 | 42.3 | 2.8 |
| vpd | 42.4 | 0.1 | 43.6 | 0.0 | 43.9 | 3.1 |
| SWC | 9.0 | 47.2 | 9.7 | 43.1 | 13.7 | 29.4 |
| ET | 7.0 | 52.5 | 4.2 | 56.4 | 0.1 | 64.8 |
| Sep Oct Nov | Dim.1 | Dim.2 | Dim.1 | Dim.2 | Dim.1 | Dim.2 |
| Tair | 35.5 | 0.4 | 34.8 | 0.8 | 37.6 | 0.4 |
| vpd | 32.6 | 0.0 | 31.8 | 0.2 | 33.5 | 0.4 |
| SWC | 10.9 | 73.6 | 11.0 | 78.4 | 13.3 | 54.4 |
| ET | 21.1 | 25.9 | 22.4 | 20.6 | 15.7 | 44.8 |






Table B2: Same as Table B1 but for DE-Gri (grassland site).

| | DE-Gri | | | | | |
|---|---|---|---|---|---|---|
| | ET_uncorr | | ET_corr | | ET_residual | |
| Dec, Jan, Feb | Dim.1 | Dim.2 | Dim.1 | Dim.2 | Dim.1 | Dim.2 |
| Tair | 29.1 | 5.6 | 29.0 | 6.1 | 35.9 | 0.6 |
| vpd | 36.8 | 6.5 | 37.3 | 6.5 | 24.7 | 24.2 |
| SWC | 3.7 | 80.9 | 3.8 | 79.8 | 8.7 | 74.8 |
| ET | 30.5 | 7.0 | 29.9 | 7.7 | 30.7 | 0.4 |
| Mar Apr May | Dim.1 | Dim.2 | Dim.1 | Dim.2 | Dim.1 | Dim.2 |
| Tair | 25.9 | 11.1 | 26.2 | 9.2 | 26.6 | 7.3 |
| vpd | 28.2 | 5.5 | 28.4 | 4.5 | 28.7 | 3.6 |
| SWC | 17.4 | 81.1 | 17.0 | 82.3 | 16.7 | 83.0 |
| ET | 28.5 | 2.3 | 28.4 | 3.9 | 28.0 | 6.2 |
| Jun Jul Aug | Dim.1 | Dim.2 | Dim.1 | Dim.2 | Dim.1 | Dim.2 |
| Tair | 32.3 | 0.2 | 31.9 | 0.3 | 31.2 | 0.5 |
| vpd | 37.6 | 0.1 | 37.1 | 0.1 | 36.8 | 0.0 |
| SWC | 9.4 | 69.0 | 9.3 | 71.5 | 9.7 | 74.0 |
| ET | 20.6 | 30.7 | 21.7 | 28.0 | 22.3 | 25.5 |
| Sep Oct Nov | Dim.1 | Dim.2 | Dim.1 | Dim.2 | Dim.1 | Dim.2 |
| Tair | 28.0 | 6.1 | 27.8 | 6.2 | 28.2 | 6.3 |
| vpd | 31.0 | 0.6 | 30.9 | 0.7 | 30.4 | 0.4 |
| SWC | 14.0 | 83.1 | 13.9 | 83.4 | 13.4 | 83.5 |
| ET | 27.1 | 10.2 | 27.4 | 9.7 | 28.0 | 9.8 |






Table B3: Same as Table B1 but for DE-Kli (crop rotation).

| | DE-Kli | | | | | |
| | ET_uncorr | | ET_corr | | ET_residual | |
| Dec, Jan, Feb | Dim.1 | Dim.2 | Dim.1 | Dim.2 | Dim.1 | Dim.2 |
|---|---|---|---|---|---|---|
| Tair | 33.9 | 5.8 | 35.0 | 5.8 | 40.1 | 0.0 |
| vpd | 38.2 | 3.1 | 38.5 | 2.7 | 27.1 | 9.0 |
| SWC | 3.2 | 76.7 | 3.3 | 74.9 | 5.0 | 89.3 |
| ET | 24.6 | 14.5 | 23.3 | 16.7 | 27.8 | 1.7 |
| Mar Apr May | Dim.1 | Dim.2 | Dim.1 | Dim.2 | Dim.1 | Dim.2 |
| Tair | 30.2 | 7.9 | 31.1 | 8.5 | 33.4 | 10.6 |
| vpd | 29.7 | 6.0 | 29.8 | 6.6 | 30.1 | 10.0 |
| SWC | 10.5 | 86.1 | 11.0 | 84.9 | 12.4 | 78.4 |
| ET | 29.6 | 0.1 | 28.1 | 0.0 | 24.0 | 1.0 |
| Jun Jul Aug | Dim.1 | Dim.2 | Dim.1 | Dim.2 | Dim.1 | Dim.2 |
| Tair | 34.9 | 0.8 | 32.9 | 5.0 | 32.8 | 4.1 |
| vpd | 39.1 | 0.1 | 37.4 | 1.5 | 37.0 | 1.1 |
| SWC | 9.1 | 77.9 | 9.5 | 89.6 | 9.0 | 89.6 |
| ET | 16.9 | 21.2 | 20.3 | 4.0 | 21.2 | 5.2 |
| Sep Oct Nov | Dim.1 | Dim.2 | Dim.1 | Dim.2 | Dim.1 | Dim.2 |
| Tair | 37.8 | 0.0 | 37.3 | 0.1 | 38.0 | 1.1 |
| vpd | 36.1 | 2.3 | 35.4 | 1.6 | 36.3 | 0.6 |
| SWC | 2.4 | 73.7 | 3.5 | 77.1 | 7.0 | 75.2 |
| ET | 23.7 | 23.9 | 23.8 | 21.3 | 18.8 | 23.1 |





Table B4: Same as Table B1 but for DE-Obe (coniferous forest).

| DE-Obe | | | | | |
|---|---|---|---|---|---|
| ET_uncorr | | ET_corr | | ET_residual | |
| Dec, Jan, Feb | | | | | |
| | Dim.1 | Dim.2 | Dim.1 | Dim.2 | Dim.1 | Dim.2 |
| Tair | 30.1 | 15.2 | 31.1 | 14.2 | 38.8 | 2.6 |
| vpd | 46.8 | 0.0 | 46.9 | 0.1 | 38.6 | 0.1 |
| SWC | 0.1 | 60.9 | 0.2 | 60.9 | 0.9 | 84.8 |
| ET | 23.0 | 23.9 | 21.7 | 24.8 | 21.7 | 12.5 |
| Mar Apr May | Dim.1 | Dim.2 | Dim.1 | Dim.2 | Dim.1 | Dim.2 |
| Tair | 33.1 | 0.3 | 33.1 | 0.3 | 34.8 | 1.0 |
| vpd | 33.5 | 0.8 | 33.1 | 0.9 | 32.6 | 1.7 |
| SWC | 2.5 | 97.1 | 2.5 | 97.2 | 3.4 | 96.5 |
| ET | 30.9 | 1.8 | 31.3 | 1.6 | 29.2 | 0.9 |
| Jun Jul Aug | Dim.1 | Dim.2 | Dim.1 | Dim.2 | Dim.1 | Dim.2 |
| Tair | 43.5 | 0.4 | 43.1 | 0.1 | 42.3 | 0.1 |
| vpd | 43.7 | 0.3 | 43.1 | 0.0 | 41.9 | 0.0 |
| SWC | 9.8 | 43.1 | 8.9 | 46.0 | 8.8 | 47.8 |
| ET | 3.0 | 56.2 | 5.0 | 53.9 | 7.1 | 52.1 |
| Sep Oct Nov | Dim.1 | Dim.2 | Dim.1 | Dim.2 | Dim.1 | Dim.2 |
| Tair | 34.1 | 0.3 | 33.5 | 1.4 | 37.6 | 1.9 |
| vpd | 37.2 | 0.8 | 35.4 | 0.4 | 31.0 | 2.9 |
| SWC | 4.2 | 81.9 | 4.7 | 86.7 | 4.1 | 81.7 |
| ET | 24.6 | 17.0 | 26.3 | 11.5 | 27.3 | 13.5 |

**Authors Contribution**

Uta Moderow did formal analysis and investigations, wrote original draft, review and editing and visualized results. Stefanie Fischer did the gap-filling and contributed to writing and visualization. Thomas Grünwald did data curation and contributed to draft writing and review. Ronald Queck contributed to visualization and review. Christian Bernhofer contributed ideas, conceptualization, writing and review.

**Competing interest**

The authors declare that they have no conflict of interest.

**Data availability**

Data are available via various platforms including FLUXNET and ICOS.



## Acknowledgement

Data was collected within the framework of CarboEurope-Integrated Project (GOCE-CT2003-505572) and ICOS (Integrated Carbon Observation System; Grant agreement ID: 211574) of the European Commission. Operation and maintenance of investigated sites would not have been possible without the contribution and engagement of our technicians. Special thanks go to Uwe Eichelmann, Udo Postel, Heiko Prasse and Markus Hehn (all TU Dresden, Faculty of Environmental Sciences, Institute of Hydrology and Meteorology, Chair of Meteorology, Germany). We thank Dr. Rico Kronenberg (TU Dresden,
Faculty of Environmental Sciences, Institute of Hydrology and Meteorology, Chair of Meteorology, Germany) for discussing results of principal component analysis.

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
