# Peer review of "Evapotranspiration at four sites representing land-use and height gradient in the Eastern Ore Mountains (Germany)"

_Hydrology and Earth System Sciences, 2020_

## Referee Comment (RC1) · Anonymous Referee #1 · 30 Jun 2020

The study focuses on assessment of different variants of ET estimates that represent different correction schemes for surface energy imbalance. Finding the proper correction scheme for this imbalance represents a long-standing unresolved scientific problem. It builds on impressive dataset of eleven years of measurements at 4 sites that also represent 3 different land uses and 3 different elevations. Extent of the yearly differences among the ET correction variants are discussed in respect to yearly sums of precipitation and within the Budyko (1974) framework. Authors evaluate the seasonal changes in ET response to driving variables using principal component analysis. Unfortunately, in contrast with the manuscript name, land-use and height gradient are not the leading topics of the manuscript and this context is barely mentioned within the whole

manuscript. Instead the authors chose to discuss the energy balance closure (EBC) fractions and its implications for LE estimates without explicitly discussing implications for sensible heat (H) fluxes. This is understandable due to the focus of the journal and the special issue. However, this approach does not allow for a thorough discussion of EBC problem, neither are the study claims supported by an independent ET estimate using modeling or other measurement technique. All three ET estimates actually represent typical EBC correction schemes, where ET_uncorr implies attribution of EBC residual solely to H flux and they can be considered as the ET uncertainty range due to the surface energy imbalance (ET_uncorr, ET_residual) and a central value (ET_corr). Therefore, these estimates should be discussed more as the ET variants than different ET estimates that suggest application of different measurement techniques. As discussed in review Mauder et al. (2020; https://doi.org/10.1007/s10546-020-00529-6), there are sites for which attribution of EBC residual almost completely to either energy flux (H or LE; here corresponding to ET_uncorr and ET_residual) shows the best match with independent estimates of energy fluxes or modelling results. Attribution of EBC residual based on bowen ratio (here ET_corr) seems to be a pragmatic solution adopted also by FLUXNET as this approach was shown to give the best results for most of the sites that focused on such evaluations. Thus, I see as the main flaw of the study that the authors focused on a research question that cannot be answered using the applied methods and this aim is not reflected in the manuscript title. Although Budyko framework is a good tool to evaluate the potential ET, it can be hardly used to validate any of the ET variants. In their recommendation for application of ET_corr authors mainly rely on existing literature. Authors provide an analysis of ET response to environmental variables with an interesting result. However, why is ET itself used as one of the driving variables in the PCA analysis is unclear to me. One of the results that DE-Obe site shows higher ET than DE-Tha positioned in lower elevation could be interesting if developed more and it could also provide results towards the focus advertised by the manuscript title. However, authors here resort to assumption (hypothesis) about interception that is not supported by the data. Authors could evaluate

e.g. differences in available energy, its partitioning into H and LE (bowen ratio), albedo or surface conductance. An impression on interception importance could be obtained by evaluation of Priestley-Taylor coefficient. But to my understanding the differences in ET of both sites are not that large and could be simply explained by differences in annual precipitation. The manuscript is written in rather loose language and this impacts also definitions in the text. Most importantly, though central to the focus of the study, description of LE_corr estimation is practically omitted in the methods. The only mention is "LE correction follows the FLUXNET procedure (FLUXNET 2017)", while all variants represent a certain correction for lack of EBC. This makes the description ambiguous as well as some other examples in the text ("LE determined as a residual of the energy balance (ET_residual)"). Text could be also more compacted as some parts of sections are repetitions. Summarizing the above, the manuscript addresses the topic relevant for the HESS audience but instead of focusing on better understanding of the processes, it assesses different EBC corrections and documents the extent of the EBC problem. This assessment could not reach a conclusive answer due to the lack of independent ET estimates. In this respect I suggest to reject the submitted manuscript.

Minor comments: I would suggest to avoid evaluation of ET correction schemes and instead adopt the ET_corr after justification based on literature review. Differences among land-use types and elevations could be than evaluated. Authors could additionally discuss runoff. What are the typical bowen ratios at the sites? Title: "representing land-use and height gradient" -> representing different land-use and elevation gradient (land-use gradient does not applicable) Short summary: "we recommend using a distinct ET estimate" What is distinct ET? "water temporally stored on plant surfaces should receive more attention" -> Relaxed language. I believe that you propose that evaporation of intercepted water should be studied more.

---

## Referee Comment (RC2) · Anonymous Referee #2 · 9 Jul 2020

The present manuscript title implies a comparison of ET depending on land-use types and elevation differences. Instead, it is more focusing on different ET estimates (which are not completely independent) for different land-use types. The work is based on long-term data sets that would allow intensive analysis with comprehensive statistics. Hypothesis with respect to e.g. evaporation from intercepted water can unfortunately not be supported by data. The authors mainly confirm the insufficiency of ET derived from the energy balance residual on different time-scales and for different land-use types. English language should be checked especially tempi (e.g. 'analysis was done' (past), but 'variance explains' (present tense)), singular and plural, usage of prepositions but also many other issues. The paper should be thoroughly re-written, analysis

should concentrate more on the land use differences instead of intra and inter-annual differences of the different ET estimates.

Specific comments (without language suggestions):

-LE derived from eddy covariance should be mentioned in the abstract

-Differences between sites should be roughly mentioned in the abstract in case title / focus stays on land-use and elevation

-Uncertainty of EC-data has to be quantified

-Check and reduce repetitions

-Order of paragraphs in chapter 3 should be changed: 1) Eddy-Covariance 2) Heat storage 3) Gap filling 4) Evapotranspiration estimates

-Gap filling data from different year with same crop: what about met conditions?

-Discussion of the results should be more precise instead of using 'was larger', 'closely related'

-Reduce chapter 4.2 (annual scale), instead focus more on seasonal scale, land-use and elevation

-Page 1, Line 23: 'latent heat of vaporization'

-Page 2, Line 56: do you intend to say that a low mountain range is better suited for such a study?

-Page 3, Line 71: 'complete years' or 'years with complete data sets'?, similar instrumentation for EC measurements?

-Table 1: 'wind components' instead of simply 'wind', 'H2O concentration' instead of 'Humidity'

-Page 10, Line 248: cite Wutzler et al, BGS, 2018

-Table 3: a bit overwhelming, the ranges as well as extremes of the EBRs could simply be mentioned in the text

-Fig. 7: unclear whether day-time half-hourly values or daily means/sums were used

-Figs. 9 + 10 might be omitted, instead concentrate on land-use/elevation differences

---

## Author Comment (AC1) · 18 Jul 2020

Dear Reviewer

We greatly appreciate your review and the time you have taken. We are thankful for your numerous helpful and encouraging comments, which include suitability of the manuscript title, missing of independent ET estimates, relevance of interception, and application of PCA as well as language aspects We agree predominantly and apologize for submitting the manuscript in its present form. As the immediate situation does not allow for the necessary thorough revision, we have decided to withdraw the manuscript.

We thank you for your constructive review.

Yours sincerely

Uta Moderow on behalf of all Co-Authors

---

## Author Comment (AC2) · 18 Jul 2020

Dear Reviewer,

We are very grateful to you for helpful comments on the manuscript. We must acknowledge that most comments are in accordance with reviewer 1 (e.g. title vs. focus of manuscript, language aspects) but also add special questions concerning exploration of land use effects, seasonal analysis and uncertainty of EC-measurements. As the immediate situation does not allow for the necessary thorough revision, we have decided to withdraw the manuscript.

We thank you for your constructive review.

Yours sincerely

Uta Moderow on behalf of all Co-Authors
* * *